# Want to Impact Physical, Technical, and Tactical Performance during Basketball Small-Sided Games in Youth Athletes? Try Differential Learning Beforehand

**DOI:** 10.3390/ijerph17249279

**Published:** 2020-12-11

**Authors:** Sogand Poureghbali, Jorge Arede, Kathrin Rehfeld, Wolfgang Schöllhorn, Nuno Leite

**Affiliations:** 1Institute of Sport Science, Otto-von-Guericke-Universität Magdeburg, 39104 Magdeburg, Germany; sogandpoureghbali@gmail.com (S.P.); kathrin.rehfeld@ovgu.de (K.R.); 2Research Centre in Sports Sciences, Health Sciences and Human Development, CIDESD, 5001-801 Vila Real, Portugal; jorge_arede@hotmail.com; 3Institute of Sport Science, Training and Movement Science, University of Mainz, 55122 Mainz, Germany; schoellw@uni-mainz.de

**Keywords:** youth sport, team sports, learning, technology, performance analysis, training load, collective behavior, exploratory behavior, functional variability

## Abstract

This study aimed to analyze the acute effect of small-sided games, based on differential learning, on the physical, technical, and positioning performance of young basketball players. Eight basketball players under 13 (U13) participated in this study. A total of eight sessions involving half-court small-sided games (4 sets × 3 min + 1 min of passive recovery) under randomly different numerical relations were performed. Before each trial, players were verbally instructed to perform the drill in one of the conditions, in random order. Pre- and post-tests were performed in the 4v4 half-court format, in each session. External load and positional data were collected via a WIMU PRO local positioning system. Individual heart rate monitoring was used to assess the internal load. Game videos also collected notational data. The results revealed that, after the intervention, the players significantly decreased the total distance covered, the peak acceleration, average speed, training impulse, and the spatial exploration index; conversely, the results confirmed an increase in the number of dribbles. Small-sided games under randomly different numerical relations imposed acute effects in distinct variables during 4v4 half-court games. However, further studies are warranted, including longer interventions and parallel-group designs, to confirm if the training-induced effects of this method are significantly better compared to other approaches.

## 1. Introduction

Despite some historical changes in sport techniques, such as the O’Brian shotput technique, the Fosbury flop, or ski-jump technique, as well as the accompanied development in performance, traditionally, in high-performance sports, it has been assumed that the maximum benefits are attained when the training stimuli are similar to competing demands [1]. In order to reproduce the physical, technical, and tactical requirements of real match play [2], coaches regularly use small-sided games (SSGs) in their training schedules. Besides their original use for the systematic development of more complex team behaviors for beginners [2,3,4], according to the common pedagogical principles from simple to complex and from easy to hard, SSGs meanwhile not only became common drills to simultaneously challenge the physical, technical, and tactical qualities in team sports [5,6], particularly in basketball [7,8], but even underwent an independent development. Three against three in street basketball or two against two in Beach volleyball grew to independent sports with its own championships. For the big games, these smaller and adjusted versions mainly intend to mimic elements of the target game and are assumed to increase individual participation of all players and teaching the basics of tactical behavior [9]. The use of SSGs in basketball practice can provide different physiological and technical–tactical stimuli to optimize the adaptations to overcome competition needs [7,10]. Thus, manipulating the game’s structure, dynamics, rules, and the numerical relationship between attackers and defenders promotes the emergence of novel game patterns and individual tactical movements [11]. Numerically unbalanced games can lead to improvements in the learning and training process by providing required flexibility on tactical solutions, as it seems to emphasize local information that players should attend to, in order to reveal goal-directed behaviors [12]. It was reported that playing with one less defender (i.e., 5v4) does not impact the team’s defensive ability to interrupt passes or shots, and playing with two less defenders (i.e., 5v3) generates significant changes in the offenses’ behavior, driving them to create more opportunities for shooting, scoring, and passing [12]. As originally intended, evidence from recent studies in soccer confirmed quantitatively that unbalanced situations induce distinct physical, technical, and tactical behaviors based on numerical inequality [13]. Indeed, in basketball, modifying SSGs in order to train the practice objective can change the environment characteristic. A growing body of research have studied the effects of manipulating the number of players [14,15], game rules [14], the size of the playing area [11], and the work-to-rest ratio [11,16] on dependent variables such as rate of perceived exertion and heart rate. However, the parameter’s interactions are still mainly treated as being additive with an underlying linear cause–effect relationship.

In contrast, non-linear aspects are an integral part of the holistic game-based teaching and learning approach, which was introduced as a deductive counterpoint to the inductive technique tactics teaching approach of that time, from early on [17,18]. By placing greater emphasis on the cognitive component and reinterpreting the teacher–student relationship in a constructivist manner, the theory of SSGs was extended by the Teaching Games for Understanding (TGfU) approach [19] and its Australian derivative, the Game Sense approach [20,21,22,23]. While the TGfU approach can be seen as being more oriented towards the development of a basic understanding of team behavior in games of beginners, the Game Sense approach focuses more on its cognitive variable stabilization in advanced stages. On the spectrum of teaching styles by Mosston [24], both approaches could be assigned to the inclusion (TGfU) and the guided discovery (Game sense) teaching style with a more dominant teacher role. Whereas in the first case the teacher provides alternative levels of difficulty for learners, in the latter the coach plans a target and leads the athlete to discover [25].

A further and first shift towards a more dominant role of the learner is described with the divergent discovery style that can be associated with the differential learning (DL) approach, where the teacher presents a problem and the learner has to find their own solution [26]. The DL approach can be coarsely characterized by giving diffuse energy to the learnings system in the form of increasing the continuous fluctuations that occur. In its most extreme version, DL is associated without repetitions and without corrections during the learning process in order to allow the learners to find their own solution [27]. While the first aspect intends to put more emphasis on the adaptation process, the second aspect supports the development of self-learning. The DL approach is considered highly nonlinear because of the constant adaptations of two stochastic signals where ideally one signal is given by the exercise to be executed and the other signal by the fluctuations of the learner’s performance. In order to achieve the optimal learning rates, resonance of the two stochastic signals is requested [27]. Consequently, a real self-organizing process is initiated, by giving noise to the system, either by the teacher or by the learner, without explicit information about the solution, forcing the system to instigate a new coordination or tactic strategy, which typically results in the emergence of more effective or more stable movement patterns [28]. Providing a wide variety of exercises extends not only the whole range of possible solutions for a specific task, but is likely to result in more adaptive behavior in complex dynamic sport environments [27] and also offers a direct influence on brain activation in a form that working memory and consolidation of learning contents are improved [29,30,31]. One aspect of differential learning can be stunted as exploratory behavior that plays a crucial role in collective sports since it is based on initiative and interaction between the permanently changing players [32,33] within a constantly altering environment [34]. With respect to DL theory, most studies analyzed the players’ performance-based acquisition rate in the accuracy [35], quality, and efficiency of specific technical movements in team sports [26,36,37,38]. To the best of our knowledge, very few studies have analyzed the effect of SSGs based on the differential learning approach in team sports [39]. According to the findings, the use of SSGs based on the differential learning approach facilitates the development of the creativity components (i.e., attempts, versatility, and originality), all the technical variables, and the most positional variables in an under 15 group compared to typical SSGs in youth football, as well as for an under 17 experiment group. The question that naturally arises is whether younger players achieve similar changes as the under 15 and 17 age groups. In the light of a report by Santos and colleagues, it is conceivable that the use of SSGs based on the differential learning approach also favored the regularity of positional behavior compared with typical SSGs, by increasing the individual variability in the under 13 soccer players [40]. However, the main object of this study was fostering exploration behavior to improve the creativity components, mainly, the originality, versatility, and attempts of movement actions. Therefore, this paper addresses the effects of the differential learning approach in SSGs on physical, technical, and tactical performance, particularly in youth U13 basketball players, so far lacking in the scientific literature. A better understanding of the effects of differential learning on different aspects of performance may help coaches to better schedule and design training tasks to improve their technical and tactical exploration.

## 2. Materials and Methods

### 2.1. Sample

Eight young under 13 (U13) male basketball players (mean age = 12.1 ± 0.4 years; mean height = 149.1 ± 6.3 cm; mean body mass = 42.2 ± 5.2 kg) participated in the study. All players took part in an average of six hours of basketball training (4 basketball sessions/week, 90 min/session) and a competitive match (regional level) per week for four years. Starting from the aspect that no information about the probability of the hypotheses is provided and no generalization of hypothesis can be made with the underlying type of study designs, regardless of the sample size, the sample size seems to be sufficient to provide reasonable indications for future studies according to Fisher’s original statistics [41,42,43]. Written informed consent was obtained from all participants and their parents before this investigation. The present study was approved by the University of Trás-os-Montes and Alto Douro (Ethics Committee (UID04045/2020) and conformed to the recommendations of the Declaration of Helsinki.

### 2.2. Study Design

A single-group, pre-and post-test design was followed for this exploratory study. Players were divided into two balanced teams (Groups A and B) and participated in eight non-consecutive training sessions. Each training session started with a 20 min session of low-intensity running, dribbling and passing drills, and dynamic stretching exercises. After the warm-up period, athletes participated in a pre-test, including a 4 vs. 4 (4v4) half-court, small-sided game (court dimensions: 14 m in length × 15 m in width) for 5 min (Figure 1). Then, they trained in half-court SSGs based on differential learning principles (same court dimensions) for 4 sets of 3 min with 1 min of passive recovery between sets (Figure 1). Before each trial, players were verbally instructed to play small-sided games in numerical unbalance (e.g., 2v4, 2v3, 2v1, 3v4, 4v3, 4v2, 3v2, etc.), in random order (Figure 1). There was a permanent exchange in ball possession, and no bonus was applied. No corrective feedback was provided during the tests and intervention. If the ball went out of play, other strategically placed balls allowed an immediate return to play. In each session, after the training intervention, players were involved in a post-test (4v4 half-court small-sided game for 5 min).

### 2.3. External and Internal Load Analyses

Physical and positional data were collected using a previously validated WIMU PRO local positioning system (Realtrack Systems, Almeria, Spain), which integrates multiple sensors registering at different sample frequencies [18]. The sampling frequency of the 3-axis accelerometer, gyroscope, and magnetometer was 100 Hz; 120 kPa for the barometer; and 18 Hz for the positioning system [18]. The following variables were calculated per minute: (a) distance covered (DC; m), (b) accelerations (Acc) and decelerations (Dec), and (c) Player load (PL) [18]. The average speed (km∙h^−1^), peak speed (km∙h^−1^), and peak acceleration (PAcc; m∙s^−2^) and deceleration (PDec; m∙s^−2^) were also calculated [18]. The instantaneous Player Load (PLn) was computed using the following Formula (1):(1)$${\mathrm{PL}}_{n}\;=\;\frac{\sqrt{{\left(X_{n}-X_{n-1}\right)}^{2}+\;{\left(Y_{n}-Y_{n-1}\right)}^{2}+\;{\left(Z_{n}-Z_{n-1}\right)}^{2}}}{100}$$

For the accumulated PL, ($\mathrm{PL}=\overset{m}{\underset{n=0}{\sum }}\left(\mathrm{PLn}\right)\;\times \;0.01$) computed and calculated per minute for further analysis [44]. Heart rate (HR) data were recorded continuously with individual HR monitors (Garmin, Soft Strap Premium, Lenetsa, KS, USA) and reported relative to the participants’ peak HR (HR peak), taken as the highest HR recorded throughout the testing and training period [45]. The HR peak zones were defined as follows: Zone 1 (50–60%), Zone 2 (60–70%), Zone 3 (70–80%), Zone 4 (80–90%), and Zone 5 (90–100%). Edward’s training impulses (TRIMP) were calculated based on the following formula [46]: TRIMP (AU) = (time spent in zone 1 × 1) + (time spent in zone 2 × 2) + (time spent in zone 3 × 3) + (time spent in zone 4 × 4) + (time spent in zone 5 × 5). Data were analyzed using commercially available software (WIMU SPRO Software; Realtrack Systems SL).

### 2.4. Notational and Positional Analyses

SSGs were recorded using two digital cameras from a fixed position located in the superior plane (3 m above) to cover the entire pitch and were positioned 45° from one of the goal lines. The digital cameras’ frequency was 25 Hz, and the resolution was 1280 pixels × 720 pixels. Individual variable behaviors were assessed through a computerized notational analysis using commercial software (LongoMatch, Version 1.3.2, Fluendo, Barcelona, Spain). Afterwards, the data were organized in a pre-prepared spreadsheet (Excel for Windows^®^). Measurements included the numbers of passes, dribbles, and made shots. In order to prepare the positional data, the positional coordinates from the players were exported from the local positioning units and computed using dedicated routines in MATLAB^®^ (MathWorks, Inc., Natick, MA, USA), according to previous data filtering guidelines [47]. The spatial exploration index (SEI) was obtained for each player by calculating his mean pitch position, computing the distance from each positioning time series to the mean position and, finally, computing the mean value from all the obtained distances [48]. The stretch index was calculated to measure the expansion or contraction of space in the longitudinal and lateral directions that a team demonstrated in the SSGs. The stretch index for the longitudinal and lateral directions was determined by calculating the mean of the distances between each player [49]. The team length represents the maximum distance between the same team players in the longitudinal direction, and the width defines the maximum distance between players in the width axis of the team at any given moment [50].

### 2.5. Statistical Analyses

Data are presented as the mean ± SD. All data were found not to significantly deviate from a normal distribution (Shapiro–Wilk test). Related sample t-tests were used to compare the SSGs (pre-and post-test) for all notational and positional variables. According to Fisher statistics, the statistical significance was set at a *p*-value ≤ 0.05. Additionally, the effect sizes (ESs) of the differences, mainly assigned to Neyman–Pearson statistics [51], were evaluated using Cohen’s “d”. The threshold values for Cohen’s “d” for the ESs statistics were 0–0.2 trivial, >0.2 to 0.6 small, >0.6 to 1.2 moderate, >1.2 to 2.0 large, and >2.0 very large [52]. A 2 × 8 repeated measures analysis of variance (ANOVA) was performed on the absolute values of all parameters to determine the main effects between the SSGs (pre-and post-test) and sessions (Sessions 1, 2, 3, 4, 5, 6, 7, and 8). Tukey’s post-hoc analysis was used to examine the differences between the groups’ SSGs and sessions. Partial eta-squared (η^2^_p_) was used as a measure of effect size, and values were interpreted as no effect (η^2^_p_ < 0.04), minimum effect (0.04 < η^2^_p_ < 0.25), moderate effect (0.25 < η^2^_p_ < 0.64), and strong effect (η^2^_p_ > 0.64) [53]. All statistical analyses were performed using SPSS software (version 24 for Windows; SPSS Inc., Chicago, IL, USA).

## 3. Results

The statistical analyses showed a significant main effect of session for DC (*p* < 0.0001; η^2^ = 0.39), PAcc (*p* < 0.001; η^2^_p_ = 0.33), AS (*p* < 0.001; η^2^_p_ = 0.34), PS (*p* < 0.001; η^2^_p_ = 0.33), and SEI (*p* < 0.0001; η^2^_p_ = 0.45) (Table 1). The post-hoc analysis revealed higher DC in Session 5 than in Sessions 1, 6, and 8 (*p* < 0.05). PAcc was higher in Session 5 than in Sessions 1 and 8 (*p* < 0.05); it was also higher in Session 7 than in Session 1 (*p* < 0.05). PDec and PL were higher in Session 5 than in Session 1 (*p* < 0.05). The AS was lower in Session 1 than in Sessions 5 and 7 (*p* < 0.05). PS was higher in Sessions 5 and 7 than in Sessions 1 and 6 (*p* < 0.05). Finally, a higher SEI occurred in Session 7 than in Sessions 1, 2, 3, 6, and 8 (*p* < 0.05). Moreover, the SEI was higher in Sessions 4 (*p* < 0.05) and 5 (*p* < 0.001) than in Session 6.

There was a significant effect of SSGs in DC (*p* < 0.0001; η^2^_p_ = 0.23), PAcc (*p* < 0.0001; η^2^_p_ = 0.21), AS (*p* < 0.001; η^2^_p_ = 0.19), TRIMP (*p* < 0.05; η^2^_p_ = 0.11), and SEI (*p* < 0.05; η^2^_p_ = 0.08) (Table 1). The post-hoc analysis revealed that a lower DC, PAcc and AS (*p* < 0.01), TRIMP, and SEI occurred in the post-test (*p* < 0.05) compared to the pre-test.

There was also a significant interaction effect (session x SSG) on DC (*p* < 0.0001; η^2^_p_ = 0.45), Acc (*p* < 0.01; η^2^_p_ = 0.31), DC (*p* < 0.01; η^2^_p_ = 0.31), PL (*p* < 0.0001; η^2^_p_ = 0.48), and SEI (*p* < 0.0001; η^2^_p_ = 0.40) (Table 1). The post-hoc analysis revealed a higher DC in Session 7 than in Sessions 1, 3, 6, and 8 (*p* < 0.05). Moreover, a lower DC occurred in Session 6 than in Sessions 4 and 5 (*p* < 0.05). A higher PL occurred in Session 5 compared to Session 1 (*p* < 0.05). A higher SEI occurred in Session 7 compared to Sessions 1, 2, 3, 6, and 8 (*p* < 0.05). Furthermore, a lower SEI occurred in Session 6 compared to Sessions 4 and 5 (*p* < 0.05).

The absolute values for the positional and notational variables in both the pre- and post-test per session are displayed in Figure 2. No statistical differences were observed for the stretch index (ES = −0.29), length (ES = 0.30), width (ES = 0.58), passes (ES = −0.89), and made shots (ES = 0.25) between the pre- and post-tests (all *p* > 0.05). Statistical analyses showed that players performed significantly more dribbles in the post-test (*p* < 0.0001) (Figure 2).

## 4. Discussion

This study aimed to analyze the acute effects of playing unbalanced SSGs based on the differential learning approach on the physical, technical, and positioning performances of adolescent basketball players. We found that players elicited a dynamic performance profile across the training sessions, SSGs (pre-and post-test), and interaction between DC and SEI. Moreover, players performed a higher number of dribbles and passes (not significantly) in the post-test compared to the pre-test values.

In the present study, the athletes explored significantly less space after the training program. Previously, a 10-week training program based on differential learning confirmed the effects of this approach, with a small increase in the spatial exploration index [39]. The SEI has been shown to be highly dependent on the environmental conditions, such as the amount of space available to play in (i.e., players tend to explore less space when they play in smaller spaces) [54]. That being said, the interpretation of spatiotemporal information after the differential learning training seems to be influenced by exposure time. The SSGs are characterized by a frequently changing scenario, resulting in interchangeable environmental information and stimulating the exploration of game/movement patterns [34]. Furthermore, in team sports, functional behavior depends on detecting possible actions or affordances by detecting the spatiotemporal (e.g., distances, velocities) and kinematic (e.g., posture) information during the game [55]. Considering the variability of information to which they are systematically exposed, athletes may find it difficult to reach optimal coupling between perception and action [56] (at an early stage), needing more time to feel freer to move through space. This idea seems to be reinforced by the fact that athletes in the post-test covered less distance, a critical aspect for players’ performance in youth basketball [57]. Because the decision-making in SSGs in basketball is substantially regulated by peripheral vision, particularly fixating on the free space around players [58], athletes may limit their displacement in order to guarantee individual and collective functional behavior. This could explain why beginners and children with little repetitive experience in the motor landscape are recommended to be taught with reduced stochastic perturbations, in order to be able to cope with perceptual-motor challenges [27].

Besides the decreased external load, subjects experienced lower internal load (i.e., TRIMP) values in the post-test. TRIMP was previously analyzed in another study involving under-15 and under-17 basketball players, aiming to examine the variation in the internal and external load promoted by playing with rule modifications, such as restricted dribbling, rather than regular rules [14]. The main results confirmed that the rule modifications promoted an increase in technical actions (e.g., the total number of passes), and concomitantly an increase in %HRmáx, RPE, and TRIMP [14]. Contrary to what was observed in this study, after participating in the differential learning training program, athletes performed a more significant number of technical actions, with a lower physiological impact. During exercise, a reduced heart rate results from an increase in the activity of the parasympathetic nervous system [59]. The involvement of DL tasks usually results in higher alpha power brain activity, which is often associated with being in a state of relaxation, and consequently with increased action of the parasympathetic nervous system [30,60]. This effect seems to be particularly higher during the first five minutes after performing a task, based on differential learning [33] and the same time interval between the present intervention and post-test. Usually, in high-intensity exercises such as SSG, the heart rate is regulated predominantly by parasympathetic control, through progressive baroreflex resetting, afferent feedback from muscle metaboreceptors, and systemic sympathoadrenal activation as the intensity increases [59]. However, differential learning may induce higher parasympathetic neural activity, which contains the arterial baroreflex, and feedback from muscle mechanoreceptors, limiting a rapid increase in heart rate at the beginning of exercise [59] and resulting in decreased overall TRIMP.

Although we observed a decrease in external and internal load in the post-test, athletes experienced a higher technical activity, i.e., performed a higher number of dribbles and passes (not significantly). In youth basketball, the most frequent and effective shots are made close to the basket and after dribbling [61,62]. Furthermore, the number of dribbles in basketball SSGs is influenced by time limitations, i.e., when athletes have less time to play, they perform a higher number of dribbles [63]. Based on prior information, it appears that athletes, after experiencing differential learning, seek more effective behaviors using dribbling. In the motor control landscape, movement variability promoted by differential learning generates greater neuromuscular [64] and neurophysiological adaptations [30]. Specifically, movement variability causes brain states in which certain regions produce electroencephalographic frequencies in the alpha- and theta-bands, which benefits short-term memory and motor learning [30]. Moreover, increased theta activity reflects the multi-sensory processing required to integrate information from different sensory modalities [30]. That being said, after having experienced differential learning, players will be able to attune motor control, which makes them feel more comfortable to perform technical actions. This seems to be confirmed by previous studies demonstrating that differential learning promoted a greater number of successful dribbles [39,40]. Moreover, effective use of dribbling in basketball occurred when the offense dribbled past the defense, with greater angular velocity values and decreased angular variability [65]. Thus, being able to detect possible actions based on spatiotemporal (e.g., distance, velocity) and kinematic (e.g., posture) information is paramount to progress [65].

The SSGs based on DL affects the emergent decision-making processes that are shaped based on many possible task states and provide a broader range of experiences. Thus, players could be more successful in using environmental information during less dynamic game situations (i.e., a 4v4 half-court situation) [66], adapting their behavior and considering greater reaction possibilities in the decision-making process [66]. According to TGFU and Game Sense approaches, players learn explicitly and cognitively how techniques fit into the game to meet the situated execution; consequently, they practice leading the tactics to gain the intended strategic behavior. In contrast, the DL training allowed the learners to find their own solutions without having to tell them explicitly what and to do and why. Just by confronting the learners with a changing environment, with a constant uncertainty, the solutions were found. Furthermore, continuous exposure to different and varied information stimuli consequently allows players to experience brain activity associated with advantageous attentional mechanisms [60], facilitating attunement with the relevant information; for example, to perform dribbling or passing.

Despite the usefulness of the present findings, the present study has some limitations that must be acknowledged. It could be possible that inclusion of a control group may allow a better understanding of the real effects of basketball SSGs based on DL principles. Moreover, studies on the effect of basketball SSGs based on DL principles with players of different ages, gender, sports, and training background are suggested. Thus, further studies are recommended to compare the effect of basketball SSGs based on DL principles with other game-based protocols (e.g., regular SSG) [36]. Finally, studies on the effect of basketball SSGs based on DL principles in different tasks are warranted.

## 5. Conclusions

To the best of our knowledge, this is the first study to report the effects of SSGs based on DL principles in youth basketball. The DL training program nourishes better learning about fundamental tactical principles related to space occupation, developing a consciousness of the environmental changes. They offer different motor challenges and changes in environmental information, generating further acute effects on game-based tasks at different levels. Hence, operating from previous motor experience tasks could be applied as a suitable strategy to induce the optimal motor control, through greater neuromuscular neurophysiological adaptations, and so make athletes feel more comfortable to perform technical actions. Furthermore, previous use of stochastic perturbations in game-based situations may influence interpretations of the spatiotemporal information during SSGs. That being said, this finding can be relevant to enrich session design, exploring distinct task sequences according to tactical development needs.

The declaration of this study as a pilot-study is based on the small sample size, which occurs in several studies [67,68]. Since we mainly rely on the original Fisher statistics, extended by the effect sizes according to Neyman–Pearson [51], there is no claim of generalizability [41,42,43,69]. Rather, we show further supportive evidence for the effectivity of an alternative training approach that is less coach but more athlete oriented and encourages self-organization within a divergent teaching/coaching style. In agreement with the Fisher’s statistics, we conclude, based on the *p* < 0.05 results, that it is worthwhile to pursue this kind of research on differential training.

The DL strategy may be especially useful when practitioners desire to vary training loads, in order to protect players who are going through developmental phases in which they may be more susceptible to injury risk (i.e., pubertal growth spurt) or generate distinct physiological adaptations; thus, considering the physical development needs. Our findings provide several suggestions for coaching and teaching pedagogies. This type of enhancement program could be easily applied in players’ training schedules. Coaches and educators may apply enrichment training to inspire the player’s disposition to discover and explore more since the environment is ideally suited for fostering exploration behavior and help to avoid the monotonous flow of constant training. It also nurtures fundamental motor skills and prepares players to read the game, as well as explore unusual technical–tactical behaviors.

## Figures and Tables

**Figure 1 ijerph-17-09279-f001:**
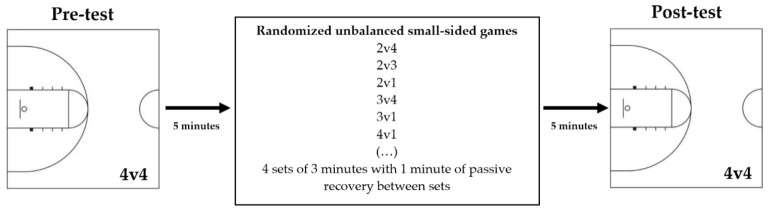
Representation of the daily intervention timeline.

**Figure 2 ijerph-17-09279-f002:**
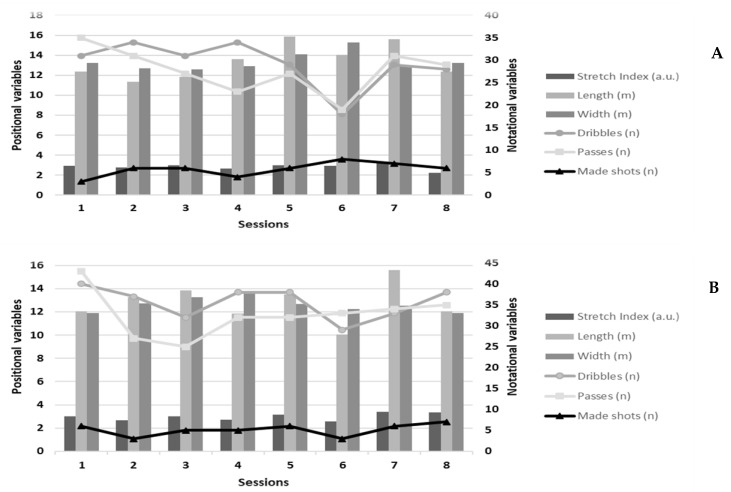
Representation of the absolute values for the positional and notational variables for each session: (**A**) = pre-test; (**B**) = post-test.

**Table 1 ijerph-17-09279-t001:** Summary of the repeated measures analyses of the performance variables.

Variable	Repeated Measures ANOVA								
	F^SESSION^	η^2^_p_	*p*	F_SSG_	η^2^_p_	*p*	F_SESSION X SSG_	η^2^_p_	*p*
Distance covered (m∙min^−1^)	5.13	0.39	0.000	16.50	0.23	0.000	6.57	0.45	0.000
Accelerations (n∙min^−1^)	1.68	0.17	0.132	0.94	0.02	0.337	3.51	0.31	0.003
Decelerations (n∙min^−1^)	1.65	0.17	0.140	0.44	0.01	0.510	3.65	0.31	0.003
Peak acceleration (m∙s^−2^)	3.96	0.33	0.001	15.02	0.21	0.000	1.57	0.16	0.163
Peak deceleration (m∙s^−2^)	2.40	0.23	0.032	6.81	0.11	0.012	2.30	0.22	0.039
Average speed (km∙h^−1^)	4.12	0.34	0.001	13.39	0.19	0.001	2.55	0.24	0.024
Peak speed (km∙h^−1^)	3.97	0.33	0.001	3.57	0.06	0.064	1.02	0.11	0.429
Player load (a.u./min.)	3.00	0.27	0.010	5.05	0.08	0.290	7.31	0.48	0.000
Peak Heart rate (bpm)	2.30	0.22	0.039	2.83	0.05	0.098	2.90	0.27	0.012
TRIMP (a.u.)	1.06	0.12	0.404	6.77	0.11	0.012	2.08	0.21	0.061
Spatial Exploration Index (a.u.)	6.63	0.45	0.000	4.85	0.08	0.032	5.23	0.40	0.000

Note: Partial eta-squared (η^2^_p_) values were interpreted as no effect (η^2^_p_ < 0.04), minimum effect (0.04 < η^2^_p_ < 0.25), moderate effect (0.25 < η^2^_p_ < 0.64), and strong effect (η^2^_p_ > 0.64).

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
