# Peer review of "Want to Impact Physical, Technical, and Tactical Performance during Basketball Small-Sided Games in Youth Athletes? Try Differential Learning Beforehand"

_ijerph, 2020, doi:10.3390/ijerph17249279_

Round 1

Reviewer 1 Report

GENERAL COMMENTS

The study examines the effects of basketball small-sided games based on differential learning. The work is meaningful and the content of the paper fits within the scope of the journal. The paper is potentially useful for enriching basketball sessions, allowing the trainers to adapt task sequences according to specific tactical demands. The manuscript is well written, and the work made by the authors is appreciated. However, I consider some improvements could be made.

SPECIFIC COMMENTS

Introduction
There is an acceptable literature review. However, the authors could improve the background. When possible, include more recently published papers from high-level scientific journals indexed in the Web of Science or Scopus databases (years 2016-2020).

Material and Methods
This section needs clarification regarding the sample size. Eight male basketball players took part in the study [Page 2, lines 76-77]. The authors should consider the derived limitations of the selected sample. What is the reason for focusing only on 12 male players? Why all of them under 13? In addition, the authors should consider the possibility of adding a table with the main characteristics of the sample.

Results
I suggest the authors to review Figure 2. Two vertical axes appear in both "A" and "B", on the left and right. Could you please clarify which of the data provided in the figure refer to each scale?

Discussion
The authors should consider including the abbreviation of differential learning (DL) the first time it appears in the text since this is a commonly used term throughout the manuscript [Page 2, line 49].

References
The references need revision. In its actual form, they do not meet the journal guidelines.

I hope the comments contribute to improving the paper.

Author Response

Comments and Suggestions for Authors

SPECIFIC COMMENTS

Introduction
There is an acceptable literature review. However, the authors could improve the background. When possible, include more recently published papers from high-level scientific journals indexed in the Web of Science or Scopus databases (years 2016-2020).

A: Some recent relevant papers have been added to the background (page 2, line 58-76)

Material and Methods
This section needs clarification regarding the sample size. Eight male basketball players took part in the study [Page 2, lines 76-77]. The authors should consider the derived limitations of the selected sample. What is the reason for focusing only on 12 male players? Why all of them under 13? In addition, the authors should consider the possibility of adding a table with the main characteristics of the sample.

A: Previous studies have emphasized the effect of DL on U15 and U17 soccer players (page3 line 101-115). A still unsolved question is whether the younger players can achieve similar changes (Page 3 line105-115).

From 12 years of age, players manage to think abstractly and develop more refined tactical group behavior during learning practice of situations closest to federated sport (González-Víllora et  al. 2015).

We claimed that there is no claim for generalization (page3, line 134-138)

Results
I suggest the authors to review Figure 2. Two vertical axes appear in both "A" and "B", on the left and right. Could you please clarify which of the data provided in the figure refer to each scale?

A: There are three vertical axes for Stretch Index, Width and Length variables, whereas dribbles, passes and made shots changes appeared by linier form.

Discussion
The authors should consider including the abbreviation of differential learning (DL) the first time it appears in the text since this is a commonly used term throughout the manuscript [Page 2, line 49].

A: The abbreviation of differential learning (DL) first time in the page 2, line 78 was corrected.

References                           
The references need revision. In its actual form, they do not meet the journal guidelines.

A: The references have been changed to the American Chemical Society (ACS) which is one of the journal guidelines.

Reviewer 2 Report

The authors provide an interesting manuscript that can add to best practice in sport coaching. There are some novel findings, but these need to be developed further within the text. The following comments are provided to assist the authors to develop the manuscript further.

Introduction:

I would suggest the authors give some attention to the literature discussing 'teaching games for understanding' (TGfU) and 'game sense coaching', as this will provide a greater rationale for the study. There is considerable literature on these topics that links with differential learning and small sided games, which will enhance the introduction and provide a better scope for analysis and discussion.

Also a greater link to learning theory may help nuance the introduction.

L70 - what is meant by 'improve their success chances'? The wording will need to be more specific here.

Materials and Methods:

There is good detail here, however I am concerned that the study is limited by the number of participants. The authors need to make some comment regarding the generalisability of the findings.

Can the authors explain any coaching interventions during the training and what the players were instructed to focus on during the activities? What I would like to know is how the coaching may have influenced learning rather than just having unbalanced sides.

Discussions:

This section needs to be split into a number of clear paragraphs that clearly distinguish between variables. This will allow the authors to add greater depth to the discussion on the findings. Currently the discussion does not provide enough detail and fails to add anything new to the current knowledge base and literature available on the topic. By separating the discussion the authors can then relate their findings back to the literature on TGfU and Game Sense coaching.

L238 - DL is used for the first time. I am assuming this is 'differential learning', but it is not clear. Could the authors please address this for consistency?

Conclusion:

Based on the developments of the introduction and discussions, the conclusion could then be updated to provide a stronger argument.

Funding: 

The authors declare no funding but on the following line acknowledge a funding source. Could this be clarified?

References:

Please check the reference list for consistency

Author Response

  1. Comments and Suggestions for Authors

Introduction:

I would suggest the authors give some attention to the literature discussing 'teaching games for understanding' (TGfU) and 'game sense coaching', as this will provide a greater rationale for the study. There is considerable literature on these topics that links with differential learning and small sided games, which will enhance the introduction and provide a better scope for analysis and discussion. Also a greater link to learning theory may help nuance the introduction.

A:The concepts of teaching games for understanding and game was added to the introduction (page 2, line 65-76) and tried to explicate clear links in whole introduction body.

L70 - what is meant by 'improve their success chances'? The wording will need to be more specific here.

A: To improve their technical and tactical exploration (page 3, line 115-118)

Materials and Methods:

There is good detail here, however I am concerned that the study is limited by the number of participants. The authors need to make some comment regarding the generalisability of the findings.

A: Material and methods ( page 3, line 124-128), Conclusion (page 8 line 347-352)

Can the authors explain any coaching interventions during the training and what the players were instructed to focus on during the activities? What I would like to know is how the coaching may have influenced learning rather than just having unbalanced sides.

A: During intervention no feedback were provided by coaches, assistants or other players. They were asked to focus on their game and their performance facing different of number of players in unexpected and random manner.

Discussions:

This section needs to be split into a number of clear paragraphs that clearly distinguish between variables. This will allow the authors to add greater depth to the discussion on the findings. Currently the discussion does not provide enough detail and fails to add anything new to the current knowledge base and literature available on the topic. By separating the discussion the authors can then relate their findings back to the literature on TGfU and Game Sense coaching.

A: At the beginning of the conclusion a summarized general perspective of study and our finding were introduced. In the second paragraph of discussion part, we presented the players displacement from positional point of view. Relevant literature and justification has been explained according to findings. The internal and external load changes have been elaborated in paragraph 3 following by players’ technical performance in paragraph 4. In paragraph 5, we represent a general discussion regarding to DL, TGFU and Game Sense approaches. In the 6 and last paragraph of discussion, the limitations of the study has been referred.

L238 - DL is used for the first time. I am assuming this is 'differential learning', but it is not clear. Could the authors please address this for consistency?

A: The abbreviation of differential learning (DL) first time in the page 2, line 78 was corrected.

Conclusion:

Based on the developments of the introduction and discussions, the conclusion could then be updated to provide a stronger argument.

 A: The conclusion section was edited and practical implication was added.

Funding: 

The authors declare no funding but on the following line acknowledge a funding source. Could this be clarified?

A: Indeed, the work was not directly funded by the strategic project of the research and development unit UID04045 / 2020, however, it is considered to be part of my activity as a member of the Research Center in Sports Sciences, Health Sciences and Human Development (CIDESD)

References:

Please check the reference list for consistency

A: The references have been changed to the American Chemical Society (ACS) which is one of the journal guidelines.

Reviewer 3 Report

This paper is very interesting. Thank you for the opportunity to review this interesting article. It surely contributes to a better knowledge of the differential learning of physical, technical, and tactical performance during basketball small-sided games in youth athletes.  However, some parts of the paper should be improved.

1.Abstract: include the sample size.

  1. It should describe the main methods or treatments applied, as well as the age of participants in the study.
  2. The general objective and specific objectives should appear at the end of the introduction. The objective should be clearly written, referring to the population, the intervention, the comparison and the results (PICO strategy)
  3. It is mandatory to inform the approval of the ethics committee; the approval number of the ethical permission.
  4. Did you apply some inclusion or exclusion criteria’s for subjects? 5.
  5. I recommend adding the exceptional reliability and validity of research tool(3-axis accelerometer, sensors...).

7.I recommend adding one or two more specific conclusion to highlight your main results of study.

Author Response

  1. Comments and Suggestions for Authors

This paper is very interesting. Thank you for the opportunity to review this interesting article. It surely contributes to a better knowledge of the differential learning of physical, technical, and tactical performance during basketball small-sided games in youth athletes.  However, some parts of the paper should be improved.

- Abstract: include the sample size.

Authors: Changed accordingly (page 1,line 16-17)

- It should describe the main methods or treatments applied, as well as the age of participants in the study.

Authors: Materials and Methods, sample and study design (page 3)

- The general objective and specific objectives should appear at the end of the introduction. The objective should be clearly written, referring to the population, the intervention, the comparison and the results (PICO strategy)

Authors: The main object was clarified (page 3, line 105-108)

- It is mandatory to inform the approval of the ethics committee; the approval number of the ethical permission.

Authors: The present study was approved by the University of Trás-os-Montes and Alto Douro (Ethics Committee (UID04045/2020) conformed to the recommendations of the Declaration of Helsinki.

- Did you apply some inclusion or exclusion criteria’s for subjects? 5.

Authors: Participants were randomly assigned

- I recommend adding the exceptional reliability and validity of research tool (3-axis accelerometer, sensors...).

Authors: Further details were included

- I recommend adding one or two more specific conclusion to highlight your main results of study.

Authors: The conclusion section was edited and practical implication was added.

Round 2

Reviewer 2 Report

The authors have done an excellent job of revising the manuscript. This will provide some good content for others working in this area.